# Dual Targeting of CDK4/6 and cMET in Metastatic Uveal Melanoma

**DOI:** 10.3390/cancers13051104

**Published:** 2021-03-04

**Authors:** Masahiro Ohara, Kengo Saito, Ken Kageyama, Mizue Terai, Hanyin Cheng, Andrew E. Aplin, Takami Sato

**Affiliations:** 1Department of Medical Oncology, Thomas Jefferson University, 1015 Walnut Street, Suite 1024, Philadelphia, PA 19107, USA; saito328@gmail.com (K.S.); kageyamaken0112@gmail.com (K.K.); Mizue.Terai@jefferson.edu (M.T.); takami.sato@jefferson.edu (T.S.); 2Department of Breast Surgery, Hiroshima General Hospital, 1-3-3 Jigozen, Hatsukaichi, Hiroshima 738-8503, Japan; 3Department of Molecular Virology, Graduate School of Medicine, Chiba University, 1-8-1 Inohana, Chuou-ku, Chiba 260-8670, Japan; 4Department of Radiology, Osaka City University, 1-4-3 Asahimachi, Abeno-ku, Osaka 545-8585, Japan; 5Department of Cancer Biology, Sidney Kimmel Cancer Center, Thomas Jefferson University, Philadelphia, PA 19107, USA; hycheng1806@gmail.com (H.C.); Andrew.Aplin@jefferson.edu (A.E.A.); 6Hematologics Inc., 3161 Elliott Ave., Suite 200, Seattle, WA 98121, USA

**Keywords:** CDK4/6 inhibitor, cMET inhibitor, metastatic uveal melanoma, combination therapy, HGF

## Abstract

**Simple Summary:**

Up to 50% of uveal melanoma patients subsequently develop metastases, for which no effective treatment has been identified. In this study, 87.5% of uveal melanoma patients’ samples were positive for phosphorylated retinoblastoma (RB), and ex vivo incubation of patients’ biopsy specimens with CDK4/6 inhibitor decreased the phosphorylation of RB. Hepatocyte growth factor (HGF), which is rich in the liver microenvironment, diminished the efficacy of CDK4/6 inhibitor. In human HGF knock-in NOD.Cg-Prkdc scid Il2rg tm1Wjl/SzJ mice, combination of CDK4/6 inhibitor and cMET inhibitor showed significant growth suppression in implanted metastatic uveal melanoma cells, compared to CDK4/6 inhibitor alone. Taken together, our preclinical study indicated that combining CDK4/6 inhibitor and cMET inhibitor would provide significant clinical benefit to patients with metastatic uveal melanoma.

**Abstract:**

Uveal melanoma (UM) is the most common cancer of the eye in adults. Up to 50% of UM patients subsequently develop metastases, especially in the liver. It has been reported that the retinoblastoma (RB) pathway is deregulated in more than 90% of UM despite the rarity of mutations in the RB1 gene itself. CDK4/6 inhibition (CDK4/6i) is a rational strategy for treatment of UM. In this report, we investigated the antiproliferative activity of a selective CDK4/6 inhibitor on metastatic UM. A CDK4/6 inhibitor suppressed UM cell lines growth in in vitro and in vivo experiments. Hepatocyte growth factor (HGF) decreased the effect of CDK4/6 inhibitor on metastatic UM cell lines. When CDK4/6i was combined with cMET inhibitor, enhanced growth suppression was observed in metastatic UM tumors grown in human-HGF knock-in xenograft mouse models. HGF is enriched in the liver and the majority of liver metastases from UM express activated forms of cMET; therefore, signaling through cMET could contribute to the resistance mechanisms against CDK4/6i, especially in UM patients with hepatic metastasis. Together, these results provide a rationale for the use of cMET inhibitor in combination with a CDK4/6 inhibitor for the treatment of metastatic UM.

## 1. Introduction

With an incidence of approximately two to eight cases per million per year in western countries, uveal melanoma (UM) remains a rare malignancy that nevertheless represents the most common primary intraocular tumor in adults [1]. Despite the development of effective local therapies, metastases occur in up to 50% of patients, predominantly in the liver [2]. The reported median survival time of metastatic UM patients ranges from only three to 12 months [3,4]. To improve the poor survival of patients with metastatic UM, development of efficacious new therapeutic approaches is needed [4,5].

More than 80% of UM have mutations in the genes GNAQ and GNA11 (collectively referred to as GNAQ/11), which encode small GTPases [6,7,8]. Oncogenic signaling as a result of GNAQ/11 mutations activates the MEK-ERK, Hippo-YAP, and PI3K/AKT/mTOR pathways [9,10,11]. Most notably, MAPK pathway inhibition with MEK inhibitors showed growth suppression of metastatic UM cell lines [12,13] and was; therefore, expected to be a viable therapeutic strategy. In a phase II trial, a MEK inhibitor, selumetinib, exhibited a 15% partial response rate in metastatic uveal melanoma patients and improved median progression-free survival (PFS), compared to a standard chemotherapy (15.9 versus 7.0 weeks, respectively). However, a subsequent phase III clinical trial (SUMIT trial), comparing selumetinib in combination with dacarbazine versus dacarbazine alone, did not meet its primary endpoint, because it showed a poor response rate and only a one-month improvement in median PFS [14].

A potential explanation for the lack of efficacy of MEK inhibition in actual patients could be a compensatory mechanism provided by the tumor microenvironment. Our group reported that one such mechanism is mediated by hepatocyte growth factor (HGF) [15,16]. We reported that HGF induced resistance to MEK inhibition in UM [17,18]. HGF is primarily produced in the liver and promotes the growth of various malignancies by increasing angiogenesis, cell motility, and cell growth, as well as by inhibiting apoptosis [19,20,21]. Prior studies indicated that HGF might play an important role in mediating UM growth and metastasis [15,16,22,23]. The receptor for HGF, cMET (mesenchymal-to-epithelial transition factor), is a plasma membrane-localized tyrosine kinase receptor that is activated by HGF [24] and expressed in human UM [16,25,26]. cMET expression was especially high in metastatic tumors (80%) [25,26]. In addition, a significant association between cMET expression and UM specific-mortality was noted [16].

Combination of MEK inhibitor with other signal blockades such as PKC inhibitor, AKT inhibitor, or cMET inhibitor are theoretically reasonable to potentially increase clinical efficacy in metastatic UM; however, MEK-based combination treatments are generally more toxic and long-term treatment would be difficult. As an alternative approach, we investigated the role of cyclin-dependent kinases (CDK) 4/6 inhibition in metastatic UM as a downstream target for signal activation related to G-protein mutations. In most UM cases, the retinoblastoma (RB) pathways are functionally inhibited although actual mutations in the *RB1* gene are rare [27,28]. As a result of cyclin D1 overexpression in about 65% of UM cases, the RB protein is constitutively hyperphosphorylated and functionally inactivated [29]. Inactivation of p16INK4a (CDKN2A), an endogenous inhibitor of CDK4, occurs frequently in primary UM (32%) and UM cell lines (50%) as a result of promoter methylation [30]. In addition to driver mutations in GNAQ/11, these characteristics of UM biology suggest that targeting CDK4/6 activity could be a valuable therapeutic strategy. Our results here indicate that CDK4/6 inhibition has a potent anti-proliferative effect on metastatic UM. We also demonstrated that HGF, rich in the tumor microenvironment in the liver, reduces the effect of CDK4/6 inhibitor in UM. The addition of cMET inhibition to the CDK4/6 inhibitor overcame the resistance mechanism induced by HGF.

## 2. Results

### 2.1. Abemaciclib Induces G1 Arrest and Decreases Cell Growth in Metastatic Uveal Melanoma Cells

To investigate the potential impact of CDK 4/6 inhibitor on metastatic UM, we examined abemaciclib, a selective CDK4/6 inhibitor, on three metastatic UM cell lines derived from UM metastases. Cells were treated with abemaciclib at concentrations ranging from 0.06 to 4.0 μmol/L for 48 h. This treatment reduced the phosphorylation and subsequent activation of RB protein, which resulted in down-regulation of the E2F target genes and decreased cyclin A2 and FOXM1 proteins as well as the mRNAs of *TOPO2* and *TK1* (Figure 1A and Appendix A). Additionally, abemaciclib treatment resulted in increased expression of cyclin D1 and CDK4 (Figure 1A), probably due to the compensatory mechanism for CDK4/6 signaling inhibition. Cell-cycle analysis demonstrated cell-cycle arrest at G0-G1 and a decreased proportion of S phase following 24 h exposure to abemaciclib (Figure 1B). Finally, abemaciclib reduced cell growth and cell viability in all three metastatic UM cell lines. The GI_50_ values for abemaciclib ranged from 0.3 to 20 μmol/L (Figure 1C,D). UM002B and UM004 were more sensitive to abemaciclib (GI_50_ = 0.43 and 0.37 μmol/L, respectively), so these cell lines were used for subsequent investigations into the mechanism of action of abemaciclib. Although it reduced the viability of the cell lines, abemaciclib treatment for 48 h did not increase cleaved PARP levels, a cellular marker for apoptosis (Figure 1E). These results indicate that abemaciclib elicits cytostasis and growth arrest, rather than apoptosis, in monolayer cultures of metastatic UM cells.

### 2.2. RB Protein Is Phosphorylated in Metastatic Uveal Melanoma Tissues and Abemaciclib Inhibits the Phosphorylation of RB in Metastatic UM Explants Obtained from Patients

Tumor biopsy samples from 16 UM patients with liver metastasis were stained with anti-phospho-RB antibodies. Staining positivity was scored as 0, negative; 1+, positive staining in <1% of tumor cells; 2+, in 1% to 10% of tumor cells; 3+, in >10% of tumor cells. We observed positive staining of phospho-RB in 87.5% of samples (two patients, 0 staining; two patients, 1+ staining; 10 patients, 2+ staining; two patients, 3+ staining). Representative images of the progressive staining grades are depicted in Figure 2A. The scores of phospho-RB were not correlated with survival rates of the patients. To test whether abemaciclib inhibits the phosphorylation of RB in metastatic UM tissues, we analyzed UM patient samples using an ex vivo treatment approach as previously described [18]. Tumor tissue pieces obtained by biopsy were treated with dimethyl sulfoxide (DMSO) or 1 μmol/L abemaciclib. As shown in Figure 2B, abemaciclib could reduce phospho-RB staining from 2+ to 0 in both patient samples. Treatment with abemaciclib inhibited the phosphorylation of RB, supporting data obtained from in vitro experiments.

### 2.3. HGF Reduces the Growth-Inhibitory Effect of Abemaciclib in Metastatic Uveal Melanoma Cells

HGF plays an important role in mediating UM growth and metastasis [15,16]. We previously reported that HGF modulated the response to MEK inhibition in UM [18,19]. To investigate the potential impact of HGF on the effects of CDK4/6 inhibitor in metastatic UM, abemaciclib treatment was administered in the presence of HGF. HGF activated AKT and ERK, and increased cell viability of UM002B and UM004 cells in a dose-dependent manner (Appendix A). HGF also reduced the effects of abemaciclib in cell viability assays (Figure 3A). These data demonstrate that HGF, rich in the liver microenvironment, can enhance the viability of metastatic UM cells that have been treated with a CDK4/6 inhibitor. To further investigate this rescue mechanism of HGF, UM002B and UM004 cells were pretreated with abemaciclib for 48 h followed by stimulation with 10 ng/mL of HGF. HGF did not affect the phosphorylation of RB induced by abemaciclib. However, it increased the phosphorylation of p70S6k induced with abemaciclib treatment (Figure 3B). p70S6k is a major downstream effector of mTORC1, which plays key roles in ribosome biogenesis and mRNA translation [31]. Our data suggest that the enhanced activation of p70S6K by HGF might compensate for the loss of cell division in abemaciclib-treated cells and thus contribute to abemaciclib resistance in metastatic UM melanoma cells. Furthermore, downstream activation of AKT and ERK by HGF was apparently not affected by abemaciclib, suggesting that a different compensatory mechanism might exist in the HGF-rich tumor microenvironment, and might play a prominent role in hepatic metastasis.

### 2.4. Merestinib Overcomes HGF-Mediated Resistance to Abemaciclib in Metastatic Uveal Melanoma Cells

To investigate whether targeting cMET could abrogate HGF-mediated resistance to abemaciclib in UM cells, we tested merestinib, a type-II ATP competitive inhibitor of the cMET tyrosine kinase [32]. In UM002B and UM004 cells, merestinib blocked HGF-induced cMET phosphorylation in a dose-dependent manner (Figure 4A). At a concentration less than 3.2 μmol/L, merestinib alone did not affect viability of either UM002B or UM004 cells (Appendix A). UM cells were treated with abemaciclib alone or in combination with merestinib, in the presence or absence of HGF. As shown in Figure 4B,C, HGF partially restored the viability and cell proliferation of abemaciclib-treated UM cells, an effect that was blocked by merestinib. The phosphorylation of cMET and downstream activation of AKT and ERK1/2, induced by HGF, were blocked by merestinib treatment. Additionally, phosphorylation of p70S6K, induced by abemaciclib and enhanced by HGF, were downregulated by merestinib (Figure 4D). Taken together, these results demonstrate that cMET inhibition reverses the HGF-mediated resistance to abemaciclib.

### 2.5. Merestinib Inhibits the HGF-Mediated Protection from Cellular Senescence in Abemaciclib-Treated Uveal Melanoma Cells

CDK4/6 inhibitor induces cytostasis that is mediated by G1 arrest and senescence. One possible mechanism of CDK4/6 inhibitor-induced senescence is through downregulation of the FOXM1 transcription factor [33]. On the other hand, HGF/cMET signaling stimulates the expression and transcriptional activity of FOXM1 via the RAS/ERK1/2, PI3K/AKT, and STAT3 signaling pathways [34,35]. We; therefore, investigated whether abemaciclib induces senescence in UM cells via downregulation of FOXM1. We also investigated whether HGF/cMET signaling interferes with this effect. As shown in Figure 5, significant increases in the percentage of SA-β-gal positive cells were found in UM002B and UM004 cells treated with abemaciclib and HGF could reduce the number of SA-β-gal positive cells. Merestinib reversed the reduction of the senescence-associated marker SA-β-gal-positivity mediated with HGF. We further analyzed the effect of HGF/cMET signaling on FOXM1 expression. Even under abemaciclib treatment, the expression of FOXM1 was increased by HGF and was accompanied by the phosphorylation of cMET and the activation of AKT and ERK1/2 (Figure 5B). HGF increased FOXM1 in a dose-dependent manner in UM004 cells treated with abemaciclib (Appendix A). Merestinib treatment reversed these changes in FOXM1 expression (Figure 5B). FOXM1 is considered to be the key element of HGF-mediated protection of UM cells from cellular senescence, and silencing of FOXM1 expression resulted in loss of HGF-mediated protection. Whereas control cells were protected from abemaciclib-induced senescence by HGF, FOXM1 knockdown resulted in loss of this protective function (Figure 5C,D). Thus, FOXM1 contributes to the HGF-mediated protection from senescence in uveal melanoma cells treated with a CDK4/6 inhibitor.

### 2.6. Abemaciclib in Combination with Merestinib Inhibits Metastatic Uveal Melanoma Tumor Growth in NSG-hHGFki Mice

Since stimulatory effects of mouse HGF on human cMET are weak, the role of HGF would be difficult to examine using traditional xenogeneic mouse models. In fact, mouse recombinant HGF did not increase the viability of UM004 cells and did not activate cMET at Tyr1349, one of the multidocking sites for recruiting downstream adaptor molecules [36]. Mouse HGF-activated signal transduction to downstream AKT and ERK in UM004 cells was either absent or very weak although cMET was phosphorylated at T1234/1235 (Appendix A). To overcome this limitation, the efficacy of the abemaciclib and merestinib combination was examined in NOD.Cg-Hgf^tm1.1(HGF)Aveo^ Prkdc^scid^ IL2rg^tm1Wjl^/J (NSG-hHGFki) mice that produce human HGF. The concentration of serum human HGF in NSG-hHGFki mice measured by ELISA (Abcam) was 12.0 ± 0.58 (S.D) ng/mL. NSG-hHGFki mice bearing implanted UM004 cells were treated with either vehicle, abemaciclib (50 mg/kg), merestinib (12 mg/kg), or a combination of abemaciclib and merestinib. Both abemaciclib and merestinib alone inhibited tumor growth, and the combination yielded greater inhibition of tumor growth than either treatment alone (Figure 6A,B). No significant changes in body weight were found between the four groups during the treatment period (Figure 6C). To elucidate further the conceptual mechanism of the abemaciclib and merestinib combination, tumors resected following seven days of treatment were analyzed by Western immunoblot and immunohistochemistry (IHC). Treatment of mice with abemaciclib resulted in inhibition of phosphorylated RB. Treatment of mice with merestinib resulted in inhibition of phospho-cMET. As expected, FOXM1 was effectively decreased by the abemaciclib and merestinib combination as compared to either treatment alone (Figure 6D,E). Assessment of the Ki67 proliferation marker by IHC confirmed that proliferative activity was suppressed in tumors treated with the abemaciclib and merestinib combination. Taken together, these results suggest that the addition of merestinib to abemaciclib enhances the suppression of UM growth in the tumor microenvironment where human HGF is produced.

## 3. Discussion

Due to the high cyclin D1 and RB expression observed in UM [28,29,30], the CDK4/6 inhibitor is a potentially attractive anticancer therapy for metastatic UM. In fact, we observed phosphorylation of RB in 87.5% of tumor biopsy specimens obtained from metastatic UM patients, consistent with a previous report [28]. In this study, we demonstrated the therapeutic potential of the CDK4/6 inhibitor, abemaciclib, in treating metastatic UM. The data presented here show that CDK4/6 inhibitor is a viable therapeutic strategy in metastatic UM. Using abemaciclib, we observed RB hypophosphorylation, cell-cycle arrest, cellular senescence, and tumor growth reduction in metastatic UM in vitro, in vivo, and ex vivo.

Despite potential efficacy demonstrated by in vitro and in vivo experiments, CDK4/6 inhibitors as single agents for cancer treatment do not appear to be very effective [37]. Although CDK4/6 inhibitor induces cell-cycle arrest accompanied by senescence, a small subpopulation of cells continually reenter the cell cycle, generating cells that escape from senescence and retain their growth capacity [38,39]. Various combination strategies have been investigated in the preclinical and clinical settings. Both PI3K/mTOR inhibitors and RAF/MEK inhibitors are being explored as potential partners of CDK4/6 inhibitor. In preclinical studies, CDK4/6 inhibitor, palbociclib, had an additive effect on cell death in pancreatic ductal carcinoma cell lines when combined with MEK or PI3K/mTOR inhibitors [40]. Combination of CDK4/6 inhibitors with RAF/MEK inhibitors is an especially promising therapeutic approach, particularly in patients with melanoma. Concurrent targeting of both CDK4/6 and MEK resulted in enhanced cell death in both BRAF- and NRAS-mutant melanoma [41]. Additionally, melanoma cell lines and tumor models that are resistant to the BRAF inhibitor vemurafenib exhibit reactivated MAPK signaling and upregulated cyclin D1, and are sensitive to abemaciclib treatment [42]. Despite conceptual merits in combining CDK4/6 inhibitor with other signal blockades, a MEK inhibitor might not be a good partner for CDK4/6 inhibitor in metastatic UM due to their class-specific and potentially serious side effects, including cardiac and eye damage.

It should also be emphasized that the tumor microenvironment, especially in the liver, might hamper the efficacy of CDK4/6 inhibitor. Emerging evidence indicates that factors from the tumor microenvironment including HGF play a significant role in the regulation of UM cells and their acquired resistance mechanisms to targeted therapies such as MEK inhibition [17,18]. This issue was not well addressed previously due to species-specificity of HGF. As reported in this paper, mouse HGF does not activate human cMET signal transduction to downstream AKT and ERK; therefore, the influence of human HGF signaling on efficacy of other signaling inhibitors cannot be tested in traditional immuno-compromised mouse models. In this regard, our mouse model is unique by testing a human “metastatic” UM cell line in NSG-hHGFki mice. As demonstrated in this report, human HGF reduces the growth-inhibitory effect of abemaciclib on metastatic UM cells in NSG-hHGFki mice.

HGF is abundant in the liver microenvironment and could contribute to the resistance mechanism to various targeted anticancer agents [17,18]. In the present study, HGF was shown to enhance mTOR activity in metastatic UM cells treated with abemaciclib. p70S6K in the mTOR pathway was stimulated with abemaciclib. mTOR activation might occur as a consequence of accumulated lysosomes and increased amino acid pools caused by the metabolic reprogramming by CDK4/6 inhibitor, and HGF could further amplify this mTOR effect by activating AKT and ERK [43,44]. In this regard, combination of CDK 4/6 inhibitor with cMET inhibitor is expected to be more efficacious.

Previous observations also suggest that cyclin D1 may inhibit TSC2, a negative regulator of mTOR, in a CDK4/6-independent manner [45]. Our results and others have shown that CDK4/6 inhibitors can lead to the accumulation of cyclinD1 to high levels [46,47]. To test whether this accumulation of cyclin D1 is associated with the activation in p70S6K, we silenced cyclin D1 expression in the cells treated with CDK4/6 inhibitor and HGF. Knockdown of cyclin D1 did not alter the activation of p70S6K (Appendix A); therefore, accumulation of cyclin D1 does not appear to be associated with the CDK4/6 inhibitor induced-mTOR activation. Together, these data highlight the possibility that HGF might enhance the resistant response to CDK4/6 inhibitor through the activation of mTOR, which might provide additional resistance to CDK4/6 inhibitor.

HGF/cMET signaling stimulates the expression and transcriptional activity of FOXM1 via its downstream pathways in pancreatic ductal adenocarcinoma and lung adenocarcinoma cells [34,35]. The activity of FOXM1, leading to suppression of cellular senescence [48], is also influenced by CDK4/6 signaling [33]. In neuroblastoma cells, CDK4/6 inhibition caused cell-cycle arrest and cellular senescence that was correlated with dose-dependent decreases in phosphorylated RB and FOXM1, respectively [49]. The present study shows that HGF increased FOXM1 and decreased the percentage of senescent SA-β-gal positive cells even during abemaciclib treatment. In FOXM1 knockdown cells, HGF did not decrease SA-β-gal positive cell rate, implying that HGF protects UM cells from CDK4/6 inhibitor induced-cellular senescence via upregulation of FOXM1, whereas cMET inhibition reverses this HGF-mediated protective effect. Consistently, the combination therapy with both abemaciclib and merestinib decreased FOXM1 to the lowest level among the four treatment groups and inhibited tumor growth significantly in hHGF-knock-in NSG mouse models.

## 4. Materials and Methods

### 4.1. Reagents

A CDK4/6 inhibitor, Abemaciclib (LY2835219), and a cMET inhibitor, merestinib (LY2801653), were provided by Eli Lilly and Company, Indianapolis, IN, USA. Abemaciclib and merestinib were dissolved in dimethyl sulfoxide (DMSO) to a 10 mmol/L concentration and stored in small aliquots at −20 °C until further use. DMSO was used as solvent control. Recombinant Human HGF (rhHGF) was purchased from R & D systems, Minneapolis, MN, USA. Antibodies against cMET, phospho-cMET (T1234/1235), AKT, phospho-AKT (S473), ERK1/2, phospho-ERK1/2, RB, phospho-RB(S780), phospho-RB(S807/811), CDK4, cyclin A2, cyclin D1, p70 S6K, phospho-p70 S6K (T389), PARP, FOXM1 and β-actin were purchased from Cell Signaling Technology, Beverly, MA, USA. An antibody against Ki67 was purchased from DAKO, Cupertino, CA, USA.

### 4.2. Cell Lines

UM001, UM002B, and UM004 were established in our laboratory and authenticated by DDC Medical, Fairfield, OH, USA. Detailed mutational analysis of these cell lines was performed by TruSeq (Illumina, San Diego, CA, USA) (Appendix A).

UM001 and UM004 cells were cultured according to the methods reported [50]. UM002B cells were maintained in MEM medium containing 15% heat-inactivated FBS and penicillin–streptomycin.

### 4.3. Short-Interfering RNA and Transfection

UM004 cells (5 × 10^5^) were seeded in 6-well plates. The next day, cells were transfected for 4 to 6 h with chemically synthesized short-interfering RNAs (siRNA) at a final concentration of 10 nmol/L using Lipofectamine RNAiMAX (Invitrogen, Carlsbad, CA, USA) according to manufacturer’s instructions. The following siRNAs were purchased from Thermo Fisher Scintific, Waltham, Massachusetts, USA.

FOXM1 siRNAs (#1: AUAGGUACCAGGUAUGAGCTG; #2: UAGGUAUUGUUGAUAGUGCA).

Cyclin D1 siRNAs (#1:UCUGGUAUCAAAAUGCUCCGG).

The non-targeting siRNA, Silencer Select Negative Control no.1 siRNA (Thermo Fisher Scientific), was used as a control.

### 4.4. MTS Assays

Cells were plated overnight at 3 × 10^3^ per well in 96-well plates and treated with 0.0625 to 16 μmol/L of abemaciclib, 0.2 μmol/L of merestinib, or 10 μg/mL of rhHGF. Following a 96-hr drug exposure, cell viability was assessed using the CellTiter 96 Aqueous One kit (Promega Corporation, Madison, WI, USA) according to the manufacturer’s instructions. Experiments were performed in triplicate. GI_50_ values were calculated using Graph Pad Prism7 software (Grapfpad Software, San Diego, CA, USA).

### 4.5. Cell Growth Assays

Cells were plated 1 × 10^5^ per well in 6-well plates. On the next day, HGF or individual inhibitors were added. Cells were cultured for an additional 7 days (medium and additives were replenished once on day 4). After fixation with 3.7% paraformaldehyde, cells were stained with 0.05% crystal violet solution for 30 min. Wells were thoroughly washed in distilled water and air-dried. Experiments were performed in triplicate. Plates were imaged by scanning while images were captured at ×200 magnification on a Nikon™ Eclipse 50i microscope (Nikon, Tokyo, Japan) with NIS-Elements D 3.1 software (Nikon).

### 4.6. Cell-Cycle Analysis

Cells were plated 1 × 10^6^ per well in 6-well plates. On the next day, cells were treated with 1 μmol/L of abemaciclib or DMSO for 24 h. Cells were detached with phosphate-buffered saline-EDTA, washed with cold PBS, and resuspended in 0.5 mL of cold PBS to make a single-cell suspension. Cells were then fixed in 4.5 mL of 70% ethanol for 2 h, followed by centrifugation at 800 g for 5 min. Cell pellets were washed with cold PBS and resuspended in 1 mL of 0.1% (*v*/*v*) Triton X-100 staining solution containing 100 ng/mL of RNase and 40 mg/mL of propidium iodide for 30 min at room temperature. Stained cells were then analyzed by flow cytometry on a BD FACSCalibur flow cytometer (BD Biosciences, San Jose, CA, USA). Experiments were performed in triplicate. Data were analyzed using FlowJo software version 9.6 (TreeStar, Ashland, OR, USA).

### 4.7. Protein Extraction and Western Immunoblotting

Cells and tumor samples were lysed on ice in M-PER Mammalian Protein Extraction Reagent (Thermo Fisher Scientific) with combined protease and phosphatase inhibitor cocktails (Thermo Fisher Scientific). Total protein was quantified using the BCA protein assay (Thermo Fisher Scientific). Equal amounts of protein (20 μg) were loaded and fractionated using a 4–20% SDS-PAGE Gel (Thermo Fisher Scientific) for Western immunoblotting assays. The protein was transferred onto PVDF membranes (Thermo Fisher Scientific). Membranes were blocked with SuperBlock T20 (TBS) blocking buffer (Thermo Fisher Scientific) and incubated with indicated primary antibodies overnight at 4 °C. Targeted proteins were detected using horseradish peroxidase-conjugated secondary antibodies followed by development using chemiluminescence substrate (Thermo Fisher Scientific). The immunoreactive bands were visualized using MSP Brand blue autoradiography film (MedSupply Partners, Atlanta, GA, USA). The quantitation of the protein bands intensity was analyzed by ImageJ software (NIH, Bethesda, Maryland, USA). The bands were normalized by β-actin.

### 4.8. Senescence Assay

Cellular senescence was assayed via measurement of senescence-associated β-galactosidase activity (SA-β-gal). Cells were plated at 1 × 10^5^ per well in 6-well plates. On the following day, HGF and/or signal inhibitors were added. Cells were cultured for an additional 6 days, and then fixed and stained overnight according to the manufacturer’s protocol (Cell Signaling #9860). Cells were imaged for SA-β-gal using a Nikon Eclipse 50i microscope (Nikon) with NIS-Elements D 3.1 software (Nikon). The percentage of SA-β-gal positive cells was determined by counting the number of positive cells present in 3 separate microscope frames and comparing to the total number of cells.

### 4.9. Xenograft Therapeutic Trials

NOD.Cg-Hgf^tm1.1(HGF)Aveo^ Prkdc^scid^ IL2rg^tm1Wjl^/J (NSG-hHGFki) mice were purchased from the Jackson Laboratory, Bar Harbor, Maine, USA and bred and kept in filter-top cages at 22 °C, 60% humidity in our animal facility. Sterile food and water were accessible *ad lib*. UM004 cells (2 × 10^6^) were injected subcutaneously in male and female 8- to 10-week-old mice and kept in the same conditions. UM004 xenograft was selected for this study because UM004 cells are sensitive to the CDK4/6 inhibitor, abemaciclib, and they grow stably in NSG-hHGFki mice. In NSG-hHGFki mice used in this experiment, human HGF is produced in any organ; therefore, the location of tumor implantation is not critical for the blocking experiments of HGF. Since consecutive accurate measurements of tumors are difficult in orthotopic liver metastasis model, we decided to use the subcutaneous implantation model to investigate the efficacy of study medications in metastatic uveal melanoma in the presence of human HGF. When tumor sizes reached approximately 100 mm^3^, mice were randomized by tumor size and subjected to each treatment. At least 5 mice per treatment group were included. Each group of mice was dosed via daily oral gavage with vehicle, abemaciclib (50 mg/kg/day), merestinib (12 mg/kg/day), or a combination of both for a total of 28 days. The abemaciclib was dissolved in 1% HEC in 20 mM phosphate buffer (pH 2.0). The merestinib was dissolved in 10%PEG400/90%(10% Captisol in H_2_O). Tumor size and body weight were measured twice and once a week, respectively. Tumor volumes were calculated using the following formula: V = (L × W^2^)/2 (L, Length; W, width). All procedures involving animals were reviewed and approved by the Institutional Animal Care and Use Committee of Thomas Jefferson University.

### 4.10. Immunohistochemistry

UM004 xenografts were excised from mice following seven days of treatment with vehicle, abemaciclib, merestinib, or in combination, fixed in formalin overnight and then embedded in paraffin. After sectioning, paraffin-embedded tissue sections were deparaffinized and antigen retrieval was accomplished using high pH conditional buffer. Sections were incubated with anti-phospho-cMET, phospho-RB, FOXM1, and Ki67 antibody overnight. On the following day, sections were incubated for 30 min in ImmPRESS AP Reagent (Vector Laboratories, Burlingame, CA, USA), followed by incubating for 5 to 10 min in ImmPact Vector Red (Vector Laboratories). Sections were counterstained with hematoxylin. Immunohistochemistry analysis was performed in three tumors from each group.

### 4.11. Patient Tissue Samples

UM metastatic tumor samples were formalin-fixed and paraffin-embedded immediately following biopsies. IHC was performed using the anti-phospho-RB (S807/811) antibody. Tissue sections were read independently by two investigators (Kengo Saito and Masahiro Ohara, Department of Medical Oncology, Thomas Jefferson University) blinded to clinical information of individual patients. The percentage of positive cells was determined by counting at least 200 cells at hot spot areas for each specimen. Four staining categories were established—0 (no detectible staining), 1+ (positive staining in <1% of tumor cells), 2+ (positive staining in 1–10%), and 3+ (positive staining in >10% of the tumor cells). Patient samples were collected under a protocol approved by the Institutional Review Board (IRB) of Thomas Jefferson University (IRB protocol number: # 11E.548).

### 4.12. Ex Vivo UM Explants

Human metastatic UM tissue was collected after obtaining patient consent for usage of their biopsy specimens for research (IRB protocol number: #02.9014R). Less than 16 h post-surgery, excess adipose and stromal tissue was removed and the tumors (explants of Patient No.1 = Ex1 and Patient No.2 = Ex2) were cut into 1 mm^3^ pieces. Vetspon absorbable hemostatic gelatin 1 cm^3^ sponges (Novartis, Basel, Switzerland) were pre-soaked in 12-well plates for 15 min at 37 °C in 500 μL of MEM medium containing 10% FBS, penicillin-streptomycin, and abemaciclib. DMSO was used as a vehicle control. To avoid concerns regarding intra-tumoral heterogeneity, up to four 1 mm^3^ pieces from different locations of the original tumor were processed per sponge per treatment condition. Samples were treated for 48 h with medium being replaced after 24 h. Tumor pieces were then formalin fixed for IHC. Mutational status of these patient specimens is as follows: Ex1, GNAQ Q209L, BAP1 wt, SF3B1 wt; Ex2, GNAQ Q209L, BAP1 wt, SF3B1 wt.

### 4.13. Statistical Analysis

Statistical analyses were performed using Graph Pad Prism7 software (Grapfpad Software, San Diego, CA, USA). The data were expressed as the mean ± standard deviation. Differences between two groups were analyzed using the two sample *t*-test with unequal variance. All of the groups were determined to be normally distributed using Shapiro–Wilk test (*P* > 0.05). One-way ANOVA analyses were performed to compare the means among multiple groups. Post-hoc Dunnett’s or Tukey’s multiple comparison tests were performed to determine statistical significance.

## 5. Conclusions

We present several lines of evidence that HGF mediates resistance to CDK4/6 inhibitor in metastatic UM cells through activation of distinct cellular mechanisms. With respect to future preclinical and clinical development, we provide proof-of-principle that targeting HGF signaling overcomes the resistance elicited by HGF. We also present evidence that combination therapy involving both CDK4/6 and cMET inhibitions is significantly more effective than single-agent approaches in hHGF knock-in mice. Based on the present study, abemaciclib in combination with merestinib should be warranted for further clinical evaluation in the treatment of metastatic UM.

## Figures and Tables

**Figure 1 cancers-13-01104-f001:**
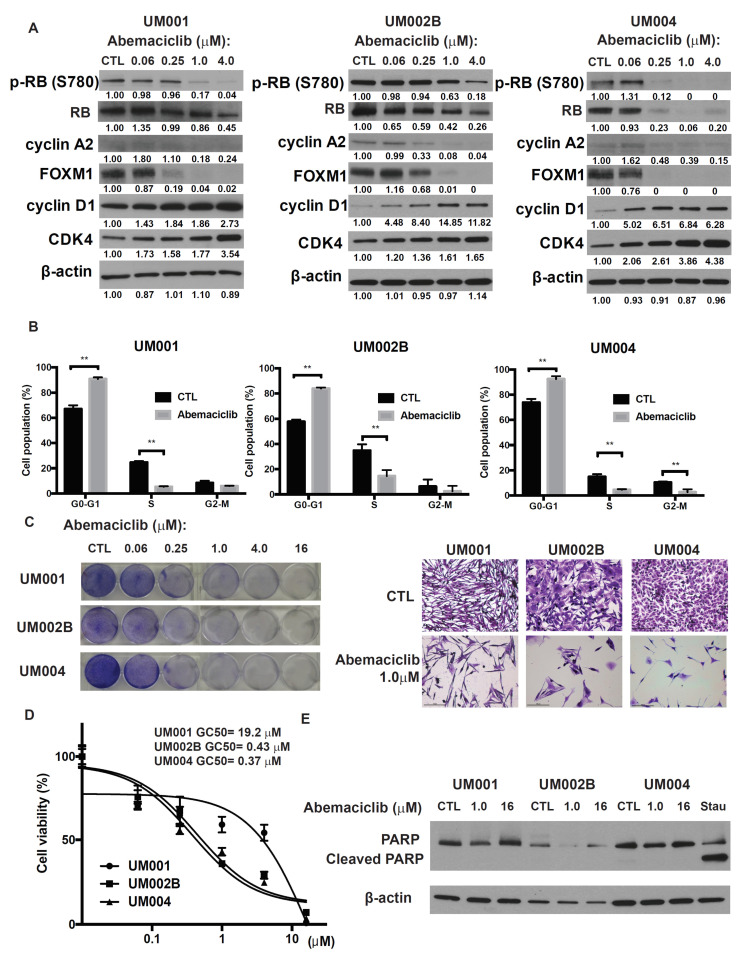
Abemaciclib induces G1 arrest and decreases cell growth in metastatic uveal melanoma cells. (**A**) UM001, UM002B, and UM004 cells were treated with DMSO or different concentrations of abemaciclib as indicated for 48 h. Cell lysates were probed with phospho-retinoblastoma (RB), total RB, cyclin A2, FOXM1, cyclin D1, CDK4, and β-actin antibodies. (**B**) Cells were treated with DMSO or 1 μM abemaciclib for 24 h. Cells were then fixed, permeabilized, and subjected to PI staining. Cell-cycle analysis was performed with FlowJo software. ** *p* < 0.01, based on the two sample *t*-test with unequal variance. (**C**) UM001, UM002B, and UM004 cells were treated with DMSO or different concentrations of abemaciclib as indicated. After seven days, cells were subjected to crystal violet staining. Representative microscopic images of the cells at ×200 magnification are shown. Scale bar, 100 μm. (**D**) UM001, UM002B, and UM004 cells were seeded in 96-well plates overnight and then treated with increasing doses of abemaciclib for 96 h. Cell viability was analyzed by the MTS assay. GI_50_ values were calculated using Graph pad Prism. (**E**) UM001, UM002B, and UM004 cells were treated with DMSO or different concentrations of abemaciclib as indicated for 48 h. Cell lysates were probed with PARP and β-actin antibodies. Stau, UM004 cells were treated with 0.5 μM of staurosporine for 3 h as a positive control.

**Figure 2 cancers-13-01104-f002:**
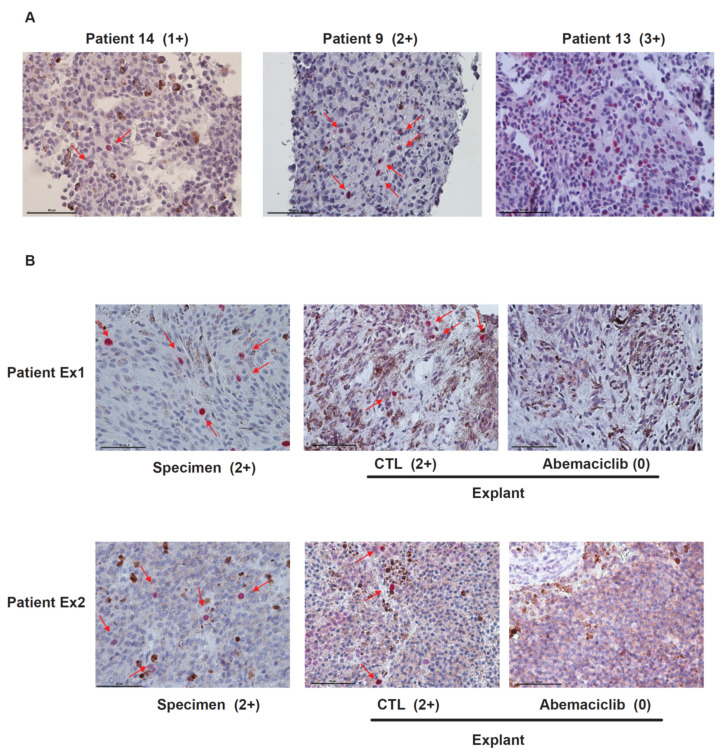
The phosphorylation of RB in metastatic uveal melanoma and the effect of abemaciclib on inhibiting phosphorylated RB in metastatic uveal melanoma (UM) explants. (**A**) Representative images (magnification, ×400) of phospho-RB staining (red color arrow) illustrating the progressively increasing staining scores for metastatic samples of uveal melanoma patients. Staining positivity was scored as 0, negative; 1+, positive staining in <1% of tumor cells; 2+, in 1% to 10% of tumor cells; 3+, in >10% of tumor cells. (**B**) Fresh UM tumors obtained from surgery were cut into approximately 1 mm^3^ pieces and plated on Vetspon absorbable hemostatic gelatin sponges for 48 h. The sponges were pre-soaked in medium containing DMSO as a vehicle control and 1 μM abemaciclib. Medium was replenished every 24 h. Surgical specimen and tumor pieces treated as above were formalin fixed and paraffin embedded. Tissue sections were stained with phospho-RB S807/811. Representative images (magnification, ×400) of phospho-RB staining (red color arrows) are shown.

**Figure 3 cancers-13-01104-f003:**
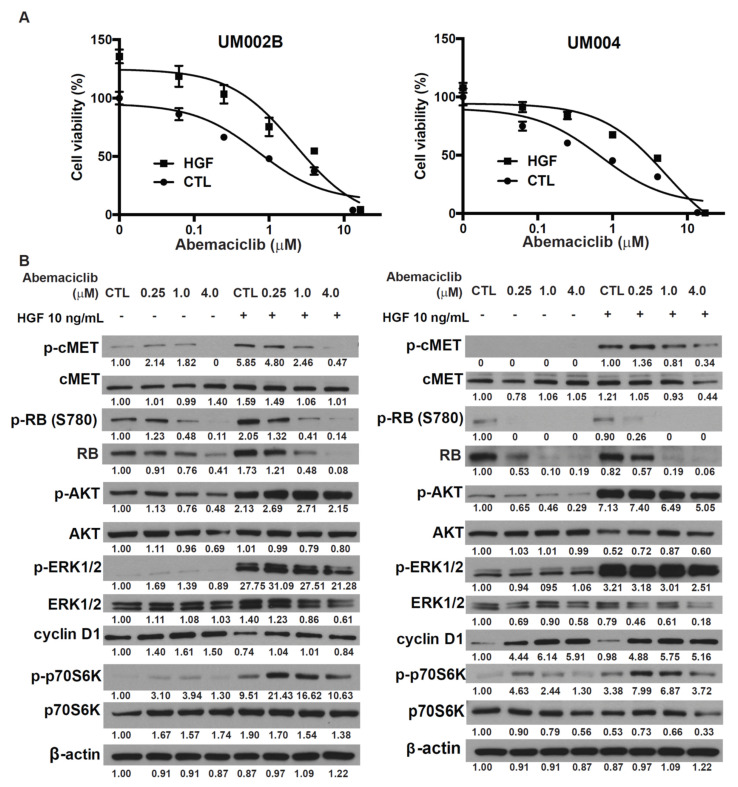
Hepatocyte growth factor (HGF) reduces the growth-inhibitory effect of abemaciclib in metastatic uveal melanoma cells. (**A**) Cells were exposed to various concentrations of abemaciclib as indicated with or without 10 ng/mL of HGF for 96 h and cell survival was assessed by the MTS assay. Data are presented as mean ± SD. (**B**) Cells were exposed to various concentrations of abemaciclib as indicated with or without 10 ng/mL of HGF for 48 h. Cells were lysed and phosphorylation of cMET, RB, AKT, ERK, and p70S6K was assessed by Western immunoblotting.

**Figure 4 cancers-13-01104-f004:**
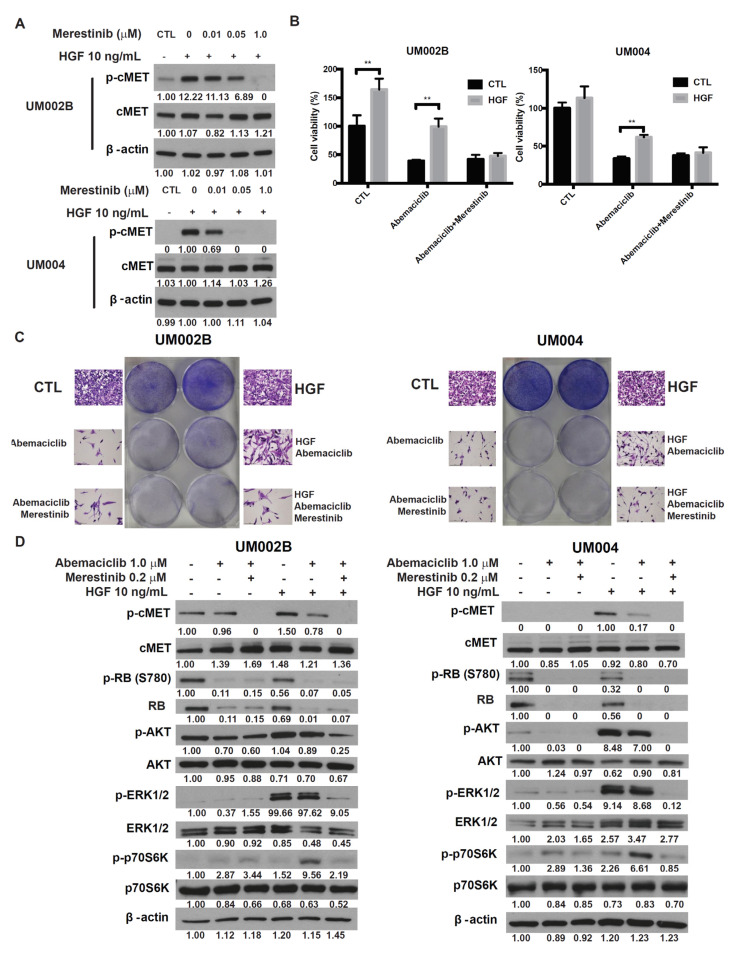
Merestinib overcomes HGF-mediated resistance to abemaciclib in metastatic uveal melanoma cells. (**A**) UM002B and UM004 cells were treated with merestinib for 4 h, followed by 10 ng/mL of HGF stimulation for 15 min. Phosphorylation of cMET was evaluated by Western immunoblotting of cell lysates with phospho-cMET antibody. Actin was used as the loading control. (**B**) UM002B and UM004 cells were treated with DMSO or 1 μM abemaciclib, in combination with 10 ng/mL of HGF and/or 0.2 μM merestinib for 4 days. Cell viability was determined by the MTS assay. Data are presented as mean ± SD. ** *p* < 0.01, based on the two sample *t*-test with unequal variance. (**C**) UM002B and UM004 cells were treated with DMSO or 1 μM abemaciclib, in combination with 10 ng/mL of HGF and/or 0.2 μM of merestinib for seven days. Cells were stained and representative images are shown. (**D**) UM002B and UM004 cells were pretreated with DMSO, 1 μM of abemaciclib, or 0.2 μM of merestinib for 48 h. Cells were then stimulated with 10 ng/mL HGF for 1 h, as indicated. Activation of cMET, RB, AKT, ERK1/2, and p70S6K was analyzed by Western immunoblotting.

**Figure 5 cancers-13-01104-f005:**
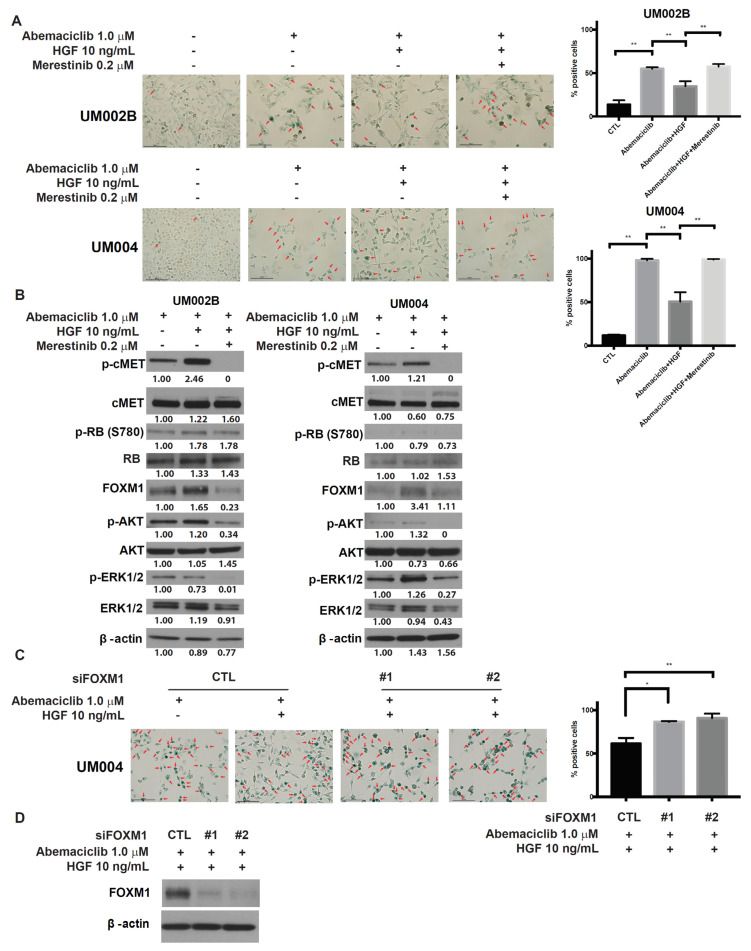
HGF prevents abemaciclib induced-cell senescence and merestinib reverses HGF effects. (**A**) UM002B and UM004 cells were treated with DMSO or 1 μM of abemaciclib, in combination with 10 ng/mL of HGF and/or 0.2 μM of merestinib for six days. Cells were fixed and stained overnight for senescence-associated β-galactosidase activity. Positive cells were shown with arrows. Data are presented as mean ± SD. *, *p* < 0.05; **, *p* < 0.01, based on Tukey analysis. (**B**) UM002B and UM004 cells were treated with 1 μM of abemaciclib, in combination with 10 ng/mL of HGF and/or 0.2 μM of merestinib for 72 h. Activation of cMET, RB, FOXM1, AKT, ERK1/2, and p70S6K was analyzed by Western immunoblotting. (**C**) UM004 cells were transfected with control or FOXM1 targeting siRNA (#1 and #2) and cultured with 1 μM of abemaciclib in the presence or absence of 10 ng/mL of HGF for six days. Following six-day treatment, cells were fixed and stained overnight for senescence-associated b-galactosidase activity. Positive cells were shown with arrows. Knockdown of FOXM1 reversed the evasion of cellular senescence mediated by HGF. (**D**) UM004 cells were transfected with control or FOXM1 targeting siRNA (#1 and #2) in the presence of 1 μM of abemaciclib and 10 ng/mL of HGF for 72 h. FOXM1 and β-actin were analyzed by Western immunoblotting.

**Figure 6 cancers-13-01104-f006:**
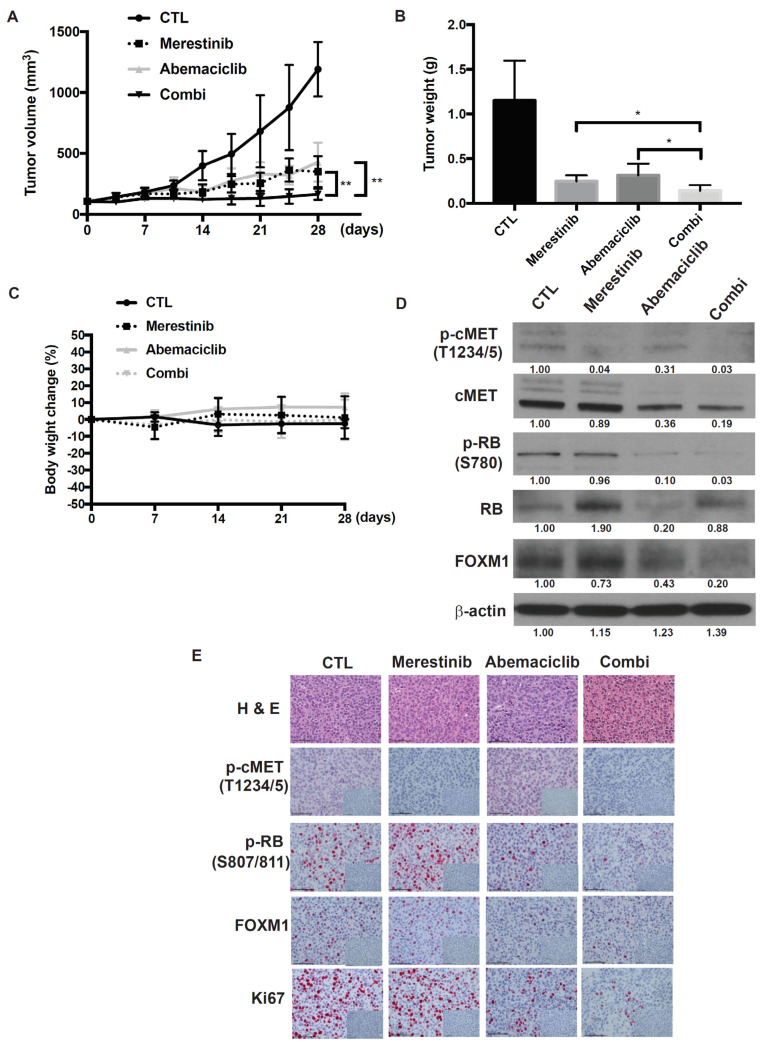
Effects of the abemaciclib and merestinib combination on a metastatic uveal melanoma xenograft tumor model in human HGF knockin mice. (**A**) Mice bearing subcutaneous UM004 tumors were dosed with vehicle (*n* = 5), 50 mg/kg abemaciclib (*n* = 5), 12 mg/kg merestinib (*n* = 5), or abemaciclib plus merestinib combination (*n* = 5) once daily for 28 days. Tumor size was measured by caliper twice weekly. (**B**) Tumor weight. Data represent mean volume ± SD. * *p* < 0.05; ** *p* < 0.01, based on Dunnett’s test. (**C**) Average percentage change in body weight after treatments. (**D**) UM004 tumors were excised from mice following seven-day treatment. Tumor lysates were prepared and analyzed by Western immunoblotting using phospho-cMET, cMET, phospho-RB, total RB, FOXM1, and β-actin. (**E**) Representative images of immunohistochemical staining for hematoxylin and eosin (H & E), IgG isotype control, phospho-cMET, phospho-RB, FOXM1, and Ki67 in UM004 tumor xenografts. Original magnification, ×400. Scale bar, 50 μm.

## Data Availability

The data presented in this study are available in this article and Supplementary Material.

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
