# Peer review of "Dual Targeting of CDK4/6 and cMET in Metastatic Uveal Melanoma"

_cancers, 2021, doi:10.3390/cancers13051104_

Round 1

Reviewer 1 Report

Abemaciclib, a cyclin-dependent kinase 4 and 6 inhibitor, has been used to treat advanced breast cancer. It was also used in the clinical trial to treat brain tumors. This manuscript gave some evidence that combining this drug and cMET inhibitor can provide clinical benefit to patients with metastatic uveal melanoma (UM). The CDK4/6 inhibitor had potent anti-proliferation effect on metastatic UM. But the high HGF in the tumor microenvironment in the liver reduces the effect of the CDK4/6 inhibitor. Therefore, cMET inhibition was needed to deal with the resistance mechanism induced by the high HGF in liver.  The results are valuable to apply abemaciclib and merestinib combination therapy in clinical trials to treat patients with metastatic UM.

The following are minor comments:

  1. In the legend for Figure 2, “()” is missing for B.
  2. The meanings of the asterisks * and ** representing significance were given on p.5. There is no need to repeat this in the figure legends for Fig.4, 6, and S1.
  3. The notations 0, 1+, 2+ and 3+ for staining positivity score were used on p.5 before they were defined in Fig. 2B.
  4. In Figure S1 the significance level notation ** should be labelled in the same manner as Fig. 1B.
  5. The tumor pictures in Fig. 6B are better shown in the figure for tumor volume.  A reader might be misled to think each group in Fig. 6B has only one tumor.

Author Response

Dear Editor,

We want to thank you and all reviewers for your constructive comments. A careful edition of the manuscript, according to the reviewer’s recommendations, was completed. We have also answered all comments.

Reviewer 1

The following are minor comments:

  1. In the legend for Figure 2, “()” is missing for B. 

Thank you for your comments. In the legend for Figure 2, “()” was added for B.

  1. The meanings of the asterisks * and ** representing significance were given on p.5. There is no need to repeat this in the figure legends for Fig.4, 6, and S1.

Thank you for your comments. In the legend for Figure 4, 6, and S1, the meanings of the asterisks * and ** were removed.

  1. The notations 0, 1+, 2+ and 3+ for staining positivity score were used on p.5 before they were defined in Fig. 2B.

Thank you for the comments. We described the scoring in page 5 as well as in the legend of Fig.2. Scores of individual patient specimens are included in the revised Figure 2.

  1. In Figure S1 the significance level notation ** should be labelled in the same manner as Fig. 1B.

In Figure S1 the significance level notation ** were changed in the same manner as Fig. 1B.

  1. The tumor pictures in Fig. 6B are better shown in the figure for tumor volume.  A reader might be misled to think each group in Fig. 6B has only one tumor.

Thank you for your comment. The tumor pictures were removed to prevent misleading the readers to think each group in Fig. 6B has only one tumor.

We would like to thank reviewer 1 for the proper and punctual corrections in order to improve the manuscript. All suggestions were considered and suggested changes were done.

Reviewer 2 Report

The manuscript “Dual targeting of CDK4/6 and cMET in metastatic uveal melanoma” by Ohara M et al. describes the potential effects of a combined therapy including CDK4/6 and cMET inhibition for metastatic uveal melanoma patients. The authors support their conclusion with in vitro, in vivo and ex vivo experiments.

The manuscript is well written, English is fine.

Few minor concerns follow:

  • Figure 1A: the authors state that abemaciclib treatment reduced RB phosphorylation. This is true for UM001 and UM002B cells; however, UM004cells show a strong reduction of RB levels, both for the total and phosphorylated protein. Consequently, reduced pRB levels could be caused by reduced levels of the protein itself. Can the authors explain why RB levels decrease in these cells?
  • Figure 3: the authors state: “Furthermore, downstream activation of AKT and ERK by HGF was apparently not affected by abemaciclib”. Looking at WB images, pAKT and pERK evidently increase in cells treated with both abemaciclib and HGF in both cells. That sounds like a contradiction. Showing all WB results also as histograms of optical density (normalized by actin beta) could make result comprehension easier.
  • Figure S3 caption: “UM002B and UM004 cells were treated with various concentrations of HGF as indicated for 96 h”. It should be merestinib, not HGF.
  • Why was siRNA silencing of FOXM1 performed in only one cell line? Previous data shown in this manuscript suggest that UM004 cells showed different results from UM002B. Using just one cell lines could be limiting.
  • Why were UM004 cells chosen for siRNA and in vivo experiments instead of UM002 cells?
  • How many mice were used for IHC analysis of tumors?
  • Statistical analysis should be better explained. How many replicates were performed for in vitro experiments? As previously suggested, histograms showing optical density of WB bands could be helpful both for understanding and showing statistical significance. Also, many figure caption state: “assuming unequal variance”. Were variance or normality of distributions analyzed to choose the best statistical test?

Author Response

We appreciate the time and effort that the reviewer has dedicated to suggesting the valuable feedback on our manuscript. We incorporated the changes provided the comments in the manuscript.

Here is a point-by-point response to the comments and concerns.

Figure 1A: the authors state that abemaciclib treatment reduced RB phosphorylation. This is true for UM001 and UM002B cells; however, UM004 cells show a strong reduction of RB levels, both for the total and phosphorylated protein. Consequently, reduced pRB levels could be caused by reduced levels of the protein itself. Can the authors explain why RB levels decrease in these cells?

Thank you for your comments. RB was reported to be the important substrate of CDK4/6 inhibitor. CDK4/6 inhibitor could decrease RB at both protein and mRNA levels. This tendency (decreased RB protein) was seen not only in UM004 cells but also in UM001 and UM002B.

Figure 3: the authors state: “Furthermore, downstream activation of AKT and ERK by HGF was apparently not affected by abemaciclib”. Looking at WB images, pAKT and pERK evidently increase in cells treated with both abemaciclib and HGF in both cells. That sounds like a contradiction.

In Figure 3, the WB band intensity of pAKT and pERK are increased in addition of HGF; however, the level of intensity was not affected by addition of abemaciclib (please compare HGF +/CTL and HGF+/ abemaciclib 0.25-4.0.). Based on this observation, we concluded that “activation of AKT and ERK by HGF was not affected by abemaciclib”. 

Showing all WB results also as histograms of optical density (normalized by actin beta) could make result comprehension easier.

The numbers for quantitation of the protein bands intensity were added in Figures of western blotting. The sentence “The quantitation of the protein bands intensity was analyzed by ImageJ software (NIH, MD, USA). The bands were normalized by b-actin.” was added in western blotting paragraph.

Figure S3 caption: “UM002B and UM004 cells were treated with various concentrations of HGF as indicated for 96 h”. It should be merestinib, not HGF.

Thank you for your comment. Figure S3 legend was changed from HGF to Merestinib.

Why was siRNA silencing of FOXM1 performed in only one cell line? Previous data shown in this manuscript suggest that UM004 cells showed different results from UM002B. Using just one cell lines could be limiting. Why were UM004 cells chosen for siRNA and in vivo experiments instead of UM002 cells?

Thank you for pointing out the important difference between UM004 and UM002B. 

In figure 3A, UM002B showed parallel decrease in cell viability with Abemaciclib between HGF+ and HGF- condition (HGF itself stimulates cell proliferation but suppressive effects of Abemaciclib are similar), while UM004 clearly demonstrated much less decrease in cell viability if HGF is added, indicating the resistance induction by HGF.  Based on this difference, UM004 cells were selected for siRNA experiments to investigate the involvement of FOXM1 for resistance mechanism.   

How many mice were used for IHC analysis of tumors?

Thank you for pointing this out. We added the sentence “Immunohistochemistry analysis was performed in three tumors from each group” in Immunohistochemistry paragraph.

Statistical analysis should be better explained.

Thank you for your comment. We added “Statistical analysis” statement to the Material and Method section. “Statistical analyses were performed using Graph Pad Prism7 software (Grapfpad Software, San Diego, California, USA). The data were expressed as the mean ± standard deviation. Differences between two groups were analyzed using the two sample t-test with unequal variance. All of the groups were determined to be normally distributed using Shapiro–Wilk test (P > 0.05). One-way ANOVA analyses were performed to compare the means among multiple groups. Post-hoc Dunnett's or Tukey's multiple comparison tests were performed to determine statistical significance.”

How many replicates were performed for in vitro experiments?

Thank you for pointing this out. We added the sentence “Experiments were performed in triplicate.” to in vitro experiment paragraph.

As previously suggested, histograms showing optical density of WB bands could be helpful both for understanding and showing statistical significance.

Thank you for this suggestion. We added the numbers for quantitation of the protein bands intensity in Figures of western blotting for showing a clear difference between control and treatment group.

Also, many figure caption state: “assuming unequal variance”. Were variance or normality of distributions analyzed to choose the best statistical test?

Thank you for your comment. We changed the sentence “the two-tailed Student t test assuming unequal variance” to “the two sample t-test with unequal variance” and described “Statistical analyses” in the material and method section. The normalities of distributions for respective data were examined by Shapiro–Wilk test. We used the two sample t-test with unequal variance. Welch's t-test performs better than Student's t-test whenever sample sizes and variances are unequal between groups, and gives the same result when sample sizes and variances are equal.